# Birds repurpose the role of drag and lift to take off and land

Diana D. Chin[1]* & David Lentink [1]*

The lift that animal wings generate to fly is typically considered a vertical force that supports weight, while drag is considered a horizontal force that opposes thrust. To determine how birds use lift and drag, here we report aerodynamic forces and kinematics of Pacific parrotlets (*Forpus coelestis*) during short, foraging flights. At takeoff they incline their wing stroke plane, which orients lift forward to accelerate and drag upward to support nearly half of their bodyweight. Upon landing, lift is oriented backward to contribute a quarter of the braking force, which reduces the aerodynamic power required to land. Wingbeat power requirements are dominated by downstrokes, while relatively inactive upstrokes cost almost no aerodynamic power. The parrotlets repurpose lift and drag during these flights with lift-to-drag ratios below two. Such low ratios are within range of proto-wings, showing how avian precursors may have relied on drag to take off with flapping wings.

[1] Department of Mechanical Engineering, Stanford University, Stanford, CA 94035, USA. *email: ddchin@alumni.stanford.edu; dlentink@stanford.edu

Like other flying animals that propel themselves, birds sustain level flight by generating net aerodynamic forces with their flapping wings that balance gravity and body drag. The net lift force on the body counters weight in the vertical direction, while net thrust counters net drag in the horizontal direction of body velocity[1,2] (Fig. 1a). In the body frame, the external work exerted on the air to generate net lift is zero, because lift acts perpendicular to the average body flight velocity and, therefore, does not oppose flight. However, lift generation induces net drag on the body, which does require aerodynamic power (drag × speed) to overcome, because drag opposes flight velocity. Consequently, aerodynamic research across engineering and biology has traditionally focused on how lift[3] is generated and can be maximized and how drag[4] can be minimized[1,2]. Although this body-centric aerodynamic force analysis has been proven particularly successful in aeronautical optimization, it is unclear how informative it is for understanding animal flight because of how their flapping wings move with respect to their body.

The notion that lift acts vertically and drag acts horizontally in level flight may become inaccurate during slow, flapping flight. While the velocities of fixed wings and rotor blades remain oriented primarily horizontally in aircraft, this is not generally true for the wings of slow-flying animals. Some hovering insects, such as dragonflies and hoverflies[5,6], utilize an inclined stroke plane that orients wing velocity, and thus drag, more vertically. Using a 2D flow simulation, Wang[6] showed how dragonflies may use an inclined stroke plane to support bodyweight with drag. Many birds also use an inclined stroke plane during slow flight[1,7–15], as do bats[1,16]. But even in the few cases where drag has been given a putative role in supporting the bodyweight of slow-flying birds[7,8,14,15,17], its contribution to weight support has never been directly measured in vivo.

The lack of quantitative studies on the role of drag in slow bird flight represents a significant limitation in our understanding of the functional constraints for the ontogeny and evolution of flapping avian flight. The wings of juvenile birds[17,18] and avian precursors with symmetrical feathers, for example, may still generate significant drag forces, despite their limited abilities to generate lift[19,20]. Furthermore, it is unclear what lift-to-drag ratio, a common measure of aerodynamic efficacy, is sufficient for avian flight over the short flight distances that pertain to evolution, ontogeny, and foraging behavior.

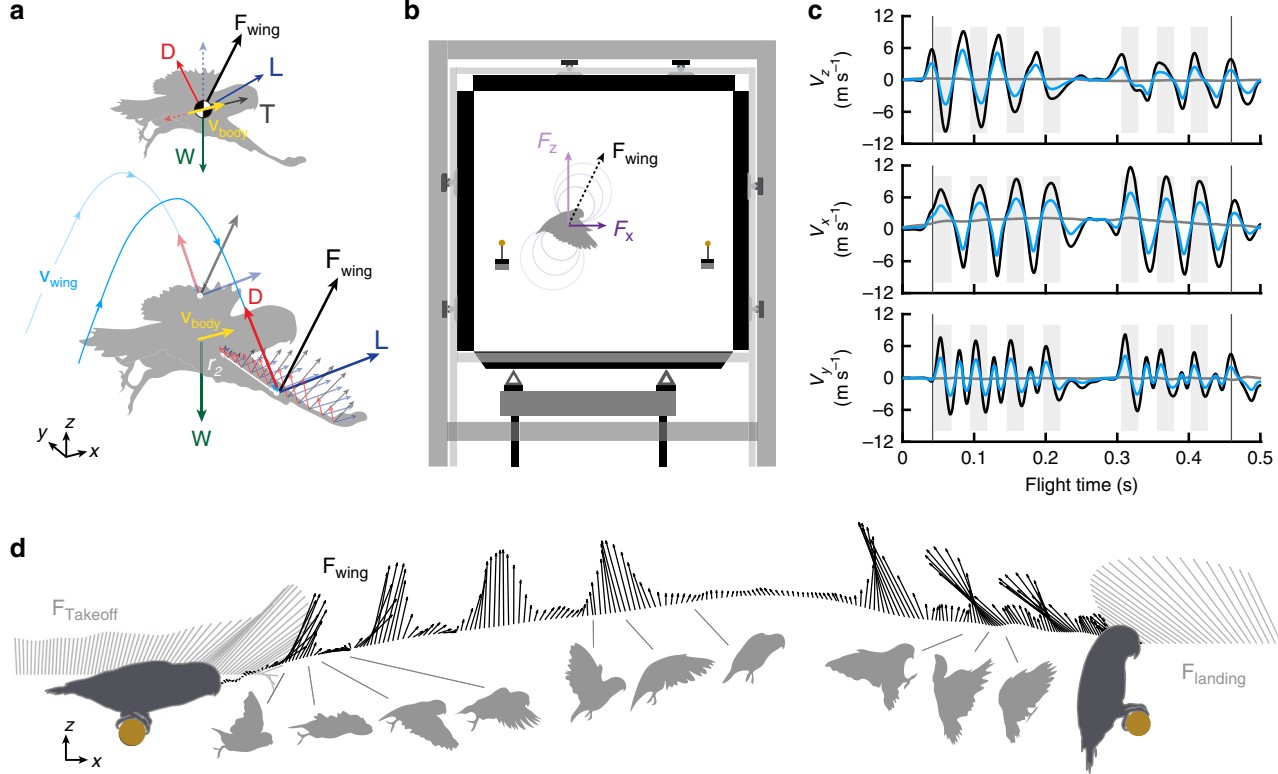

**Fig. 1** The aerodynamic force platform enables direct measurements of lift and drag components. **a** During steady forward flight, total lift **L** (dashed blue arrow) counters bodyweight **W** (solid green arrow) in the vertical direction, and total drag **D** (dashed red arrow) is countered by net thrust **T** (solid dark gray arrow) in the horizontal direction of body velocity (yellow). However, during slow flapping flight, the total lift **L** (solid blue arrow) and total drag **D** (solid red arrow) vectors generated by an individual wing are directed differently, because wing velocity $\mathbf{v}_{wing}$ does not align with body velocity $\mathbf{v}_{body}$, as shown for a bird's first downstroke after takeoff. **L** and **D** effectively act at each wing's radius of gyration $r_2$, so we base $\mathbf{v}_{wing}$ on the wing's velocity at $r_2$ (see the Methods section). Together, lift and drag make up the total force generated by the wing, $\mathbf{F}_{wing}$. **b** The net force from both wings can be decomposed into net horizontal $F_x$ and vertical $F_z$ components, both of which are directly measured in a new aerodynamic force platform (AFP). During a representative flight between two instrumented perches in the AFP (bird avatar enlarged 2× for clarity), each wing has an effective vertical velocity $V_z$, horizontal velocity $V_x$, and lateral velocity $V_y$ (**c**) at the wingtip (black) and at $r_2$ (radius of gyration; light blue). These wing velocities are governed primarily by flapping kinematics, rather than the bird's body velocity (gray). Vertical lines denote takeoff and landing, and gray-shaded regions show downstrokes. **d** We synchronize our AFP measurements with high-speed kinematics to show how the net 2D aerodynamic force vector varies in magnitude and orientation along the bird's trajectory (the tracked eye) during a representative flight. Note that 2D perch forces ($\mathbf{F}_{takeoff}$ and $\mathbf{F}_{landing}$) are simply plotted along a straight horizontal line at the same rate as the aerodynamic forces.

Here, we present in vivo aerodynamic force and kinematics measurements for perch-to-perch flights made by five Pacific parrotlets (*Forpus coelestis*). These parrotlets use an inclined stroke plane and high angles of attack up to 60° during takeoff[7]. To determine lift and drag forces resulting from these wing kinematics, we designed a new aerodynamic force platform (AFP)[21]. The AFP uses instrumented force plates on the floor and ceiling of a flight chamber to measure in vivo aerodynamic vertical forces, and similar plates that form the front and back walls to measure horizontal forces, all at 2000 Hz (Fig. 1b). To simulate foraging behavior, we provided the parrotlets with seed rewards after each flight between two instrumented perches (also recording at 2000 Hz), which were set 80 cm apart—a distance typical of foraging flights made by small, arboreal birds[22,23]. We recorded 3D kinematics at 1000 fps using five high-speed cameras, which revealed how the average velocities of the parrotlets' center of gravity ($|\mathbf{V}| = 1.70 \pm 0.16$ m s$^{-1}$) are much smaller than those of their wings (Fig. 1c), which beat at ~20 Hz. As a result, their wing velocity distributions (Fig. 1a) are determined primarily by wing flapping. The stroke-averaged Reynolds number at the radius of gyration (the second moment of wing area[24] $r_2$) is ~10,000, and reaches up to ~20,000 midstroke at the wingtip. The slow flight speeds enable us to neglect the minimal aerodynamic force contributions from the body and tail ($|\mathbf{F}_{body} + \mathbf{F}_{tail}| < 1\%$ bodyweight, see the Methods section), so measured forces can be attributed to lift and drag generated by the wings.

We were able to directly calculate lift and drag by combining our synchronized force and wing kinematics measurements (Fig. 1d). The total force on each wing $\mathbf{F}_{wing}$ can be decomposed into its Cartesian components ($F_x$, $F_y$, $F_z$) or into its drag and lift components (Fig. 2a):

$$\mathbf{F}_{wing} = \begin{pmatrix} F_x \\ F_y \\ F_z \end{pmatrix} = D \begin{pmatrix} e_{D,x} \\ e_{D,y} \\ e_{D,z} \end{pmatrix} + L \begin{pmatrix} e_{L,x} \\ e_{L,y} \\ e_{L,z} \end{pmatrix}, \quad (1)$$

where $D$ and $<e_{D,x}, e_{D,y}, e_{D,z}>$ are the magnitude and direction of drag, and $L$ and $<e_{L,x}, e_{L,y}, e_{L,z}>$ are the magnitude and direction of lift. The directions of lift and drag are determined from the direction of the wing velocity $\mathbf{v}_{wing}$ and wing radius $\mathbf{r}$; by definition, drag opposes the direction of $\mathbf{v}_{wing}$, while lift acts orthogonal to it. To uniquely determine the direction of lift, we make the reasonable assumption that $\mathbf{F}_{wing}$ acts perpendicular to $\mathbf{r}$ because it is predominately a pressure force at the Reynolds numbers associated with these flights (~10$^4$). This is particularly true for the high angles-of-attack used by the parrotlets[7], for which pressure-based profile drag dominates over friction-based profile drag[25]. We estimate that skin friction reaches a force magnitude of at most 1% bodyweight on each wing (see Methods), and therefore does not significantly alter the direction of the total aerodynamic force on the wing. This means that lift will also be perpendicular to $\mathbf{r}$ (see Methods for details). Equation 1 thus gives three scalar equations to solve for the three remaining unknowns: $D$, $L$, and the lateral force $F_y$. This governing set of equations cannot be solved with kinematics alone; only by combining our kinematic measurements with our measurements of the net vertical force ($F_z$) and horizontal force ($F_x$) are we able to solve for the magnitudes of lift and drag. We were thus able to

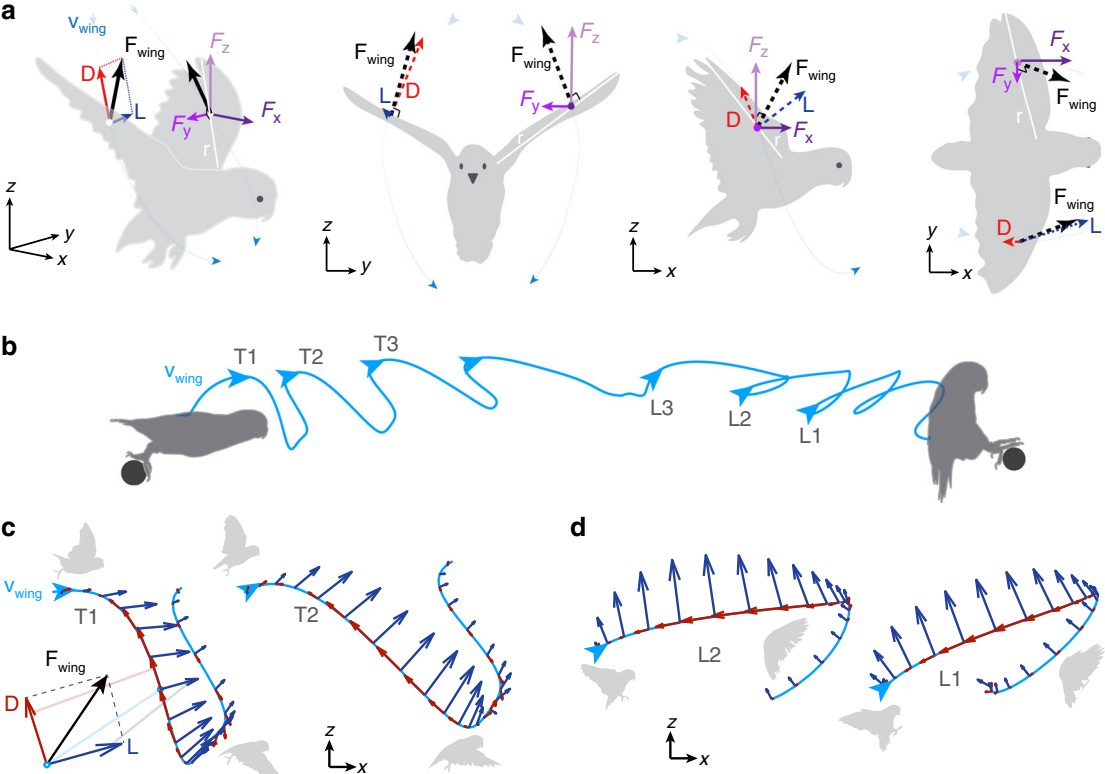

**Fig. 2** The roles of lift and drag depend on wingbeat kinematics. **a** The aerodynamic force on a wing $\mathbf{F}_{wing}$ can be decomposed into its 3D Cartesian components ($F_x$, $F_y$, $F_z$) or into orthogonal lift **L** and drag **D** components, as shown (from left to right) in the isometric, front, side, and top–down views of a parrotlet. The orientation of the wing is determined by the wing radius vector **r**. **b** As shown by the wing's $r_2$ trajectory in the sagittal (x–z) plane during a representative flight, birds initiate flights with a heavily inclined stroke plane, which gradually levels out and then pitches backward before landing (arrows denote starts of wingbeats). As a result, net drag forces are oriented more vertically during takeoff wingbeats (T1, T2, T3) (**c**), and more horizontally during landing wingbeats (L3, L2, L1) (**d**). Net lift forces have a significant forward (horizontal) component during initial wingbeats, and a backwards component prior to landing. The lengths of the reference frame axes represent 1 bodyweight for scale (except the inset in **c** is 2× magnified).

quantify the role of lift and drag from takeoff to landing for, to our knowledge, the first time.

In this work, we show how parrotlets direct drag to support bodyweight during takeoff for short, perch-to-perch flights. We also show how, during landing, they use lift to reduce the aerodynamic power cost of braking. Finally, we discuss how avian precursors may have similarly utilized high drag to support their bodyweight in order to take off with aerodynamically inefficient wings.

## Results

**Wing kinematics direct drag upwards and then lift backwards**. The first wingbeats after takeoff sweep out a heavily inclined stroke plane (Fig. 2b). Drag, which opposes wing velocity, is directed mostly up and slightly backwards, while lift is directed up and forward (Fig. 2c). As the bird continues accelerating, its stroke plane levels out and then gradually pitches backwards before landing. This causes drag to be directed more horizontally to slow down the bird, while lift is directed up and back for both weight support and deceleration (Fig. 2d) (Supplementary Movie 1).

**Lift and drag provide both weight support and braking forces**. The net aerodynamic forces produced by the birds tend to have larger vertical components for supporting bodyweight during slow flight. However, horizontal forces in the birds' flight direction are still comparable in magnitude, especially when a bird accelerates after takeoff and brakes before landing (Fig. 3a). In decomposing the net force based on the wing's velocity, we find that lift forces tend to be larger than drag forces, but both contribute substantially throughout each flight (Fig. 3b). In the vertical direction, both lift and drag contribute significantly to weight support during initial wingbeats due to the inclined stroke plane angle (Fig. 3c, e). Lift then continues to provide weight support throughout the flight, while drag contributions decrease and eventually become negative as the stroke plane tilts backwards. In the horizontal direction, drag opposes the flight direction of the bird (Fig. 3d, f). Lift initially opposes drag to accelerate the bird forward during the first few wingbeats, and then assists drag to decelerate the bird before landing.

Aerodynamic force is primarily generated during downstrokes (Fig. 3). While downstroke lift contributes the most weight support, downstroke drag also provides a significant proportion (up to 40%) of the total weight support during the first three wingbeats (T1, T2, T3) (Fig. 3e, g). Compared with downstroke drag, upstroke lift contributes less vertical force during these initial wingbeats, but more prior to landing (L3, L2, L1) (Fig. 3e, g). In terms of horizontal forces, most of the bird's acceleration during takeoff is provided by downstroke lift, although about a quarter of the forward thrust during the first wingbeat (T1) results from upstroke lift (Fig. 3f, h). Downstroke drag opposes forward motion of the bird throughout each flight. To decelerate before landing, downstroke drag is supplemented by lift forces during the final two wingbeats (L2, L1)—lift provides over a third of the total braking force during the final wingbeat (Fig. 3f, h).

**Drag increases total aerodynamic force at a cost**. The instantaneous aerodynamic power requirement varies significantly throughout each wingbeat. Notably, the instantaneous pectoralis mass-specific power briefly exceeds over $400 \, W \, kg^{-1}$ at mid-downstroke (Fig. 4a). The aerodynamic power, which we define as the rate of external work exerted on the air, is calculated as $P_{aero} = 2(\mathbf{F}_{wing} \cdot \mathbf{v}_3)$, where $\mathbf{v}_3$ is the wing's velocity at its 3rd moment of area $r_3$ (see the Methods section). During most of the wingbeat, drag force on the wing opposes its motion, requiring

positive power. However, as the wing changes direction during stroke reversal, the net force on the wing can act in the same direction as its velocity, resulting briefly in negative power. While some computational fluid dynamic (CFD) studies of flapping insect[26,27] and hummingbird[28] flight reported only positive power throughout a wingbeat, other insect studies using CFD[29,30] and robotic models[31], as well as a recent hummingbird CFD study[32], have reported similar small, negative power dips during the start and end of upstroke (Fig. 4a). These small negative power results during stroke reversal may result from the local flow field during wing-wake interactions[31]. It is also possible that wing rotation effects may account for some of the negative power, but calculating rotational power would require quantification of net torques that have never been measured before in vivo. However, compared with the translational aerodynamic power that we do measure, we expect rotational power to be relatively low, especially given the parrotlets' large stroke amplitudes (142 ± 9°)[31,33]. We therefore assume that the total aerodynamic power $P_{aero}$ can be well approximated as described above, based on the translational component of the aerodynamic power:

$$P_{aero} = P_{aero,trans} + P_{aero,rot} = 2(\mathbf{F}_{wing} \cdot \mathbf{v} + \mathbf{T}_{wing} \cdot \boldsymbol{\omega}) \approx 2(\mathbf{F}_{wing} \cdot \mathbf{v}), \quad (2)$$

where $\mathbf{T}_{wing}$ is the aerodynamic torque on the wing, and $\boldsymbol{\omega}$ is the angular velocity of the wing. Comparing stroke-averaged power (Fig. 4b), we find that upstroke requirements are minimal because upstroke drag forces are minimal (Fig. 3e, f). As a result, the downstroke is responsible for nearly all of the aerodynamic power required during each wingbeat. These power requirements tend to increase over the course of each flight (Fig. 4b) due to increasing drag forces.

To evaluate wing efficacy during these flights, we first evaluate instantaneous wing lift coefficients $C_L$ and drag coefficients $C_D$ during mid-downstroke, when the wing produces maximum net force (Fig. 4c; see Methods). We then use these coefficients to calculate the resulting power factor (PF = $C_L^{1.5}/C_D$), which is a measure of endurance efficiency[34], and the lift-to-drag ratio ($C_L/C_D$). The instantaneous power factors reach values close to 3 except during the last wingbeat, when they drop down to ~2 (Fig. 4d). The instantaneous lift-to-drag ratios remain between 1 and 2 (Fig. 4e).

## Discussion

The repurposing of lift and drag quantified in this study holds several important implications for our understanding of short, perch-to-perch flights. Our detailed aerodynamic power measurements show how these generalist birds are able to recruit high levels of pectoralis power during downstrokes while limiting the aerodynamic power needed during upstrokes. By using lift to supplement braking forces during landing, the birds are also able to reduce the total aerodynamic power required during their final wingbeats. On the other hand, using drag to support weight during takeoff increases power costs, which suggests that the birds prioritize maximizing total aerodynamic force generation during initial wingbeats.

The pectoralis mass-specific aerodynamic power output levels (Fig. 4b) reach downstroke averages of over $200–400 \, W \, kg^{-1}$. These stroke-averaged levels are similar to maximum wingbeat-averaged levels calculated from pectoralis stress and strain during burst escape flight in passerine birds[35]. Upstrokes, on the other hand, are significantly less active than downstrokes (Fig. 3), especially near the middle of each flight. As a result, the upstroke-averaged specific power requirements remain under $60 \, W \, kg^{-1}$, and average to less than $1 \, W \, kg^{-1}$ during T3 (Fig. 4b). When averaged over full wingbeats, the specific power for the parrotlet flights ($211 ± 69 \, W \, kg^{-1}$, Fig. 4b) falls near values reported for other slow-flying birds; these previous wingbeat-averaged values

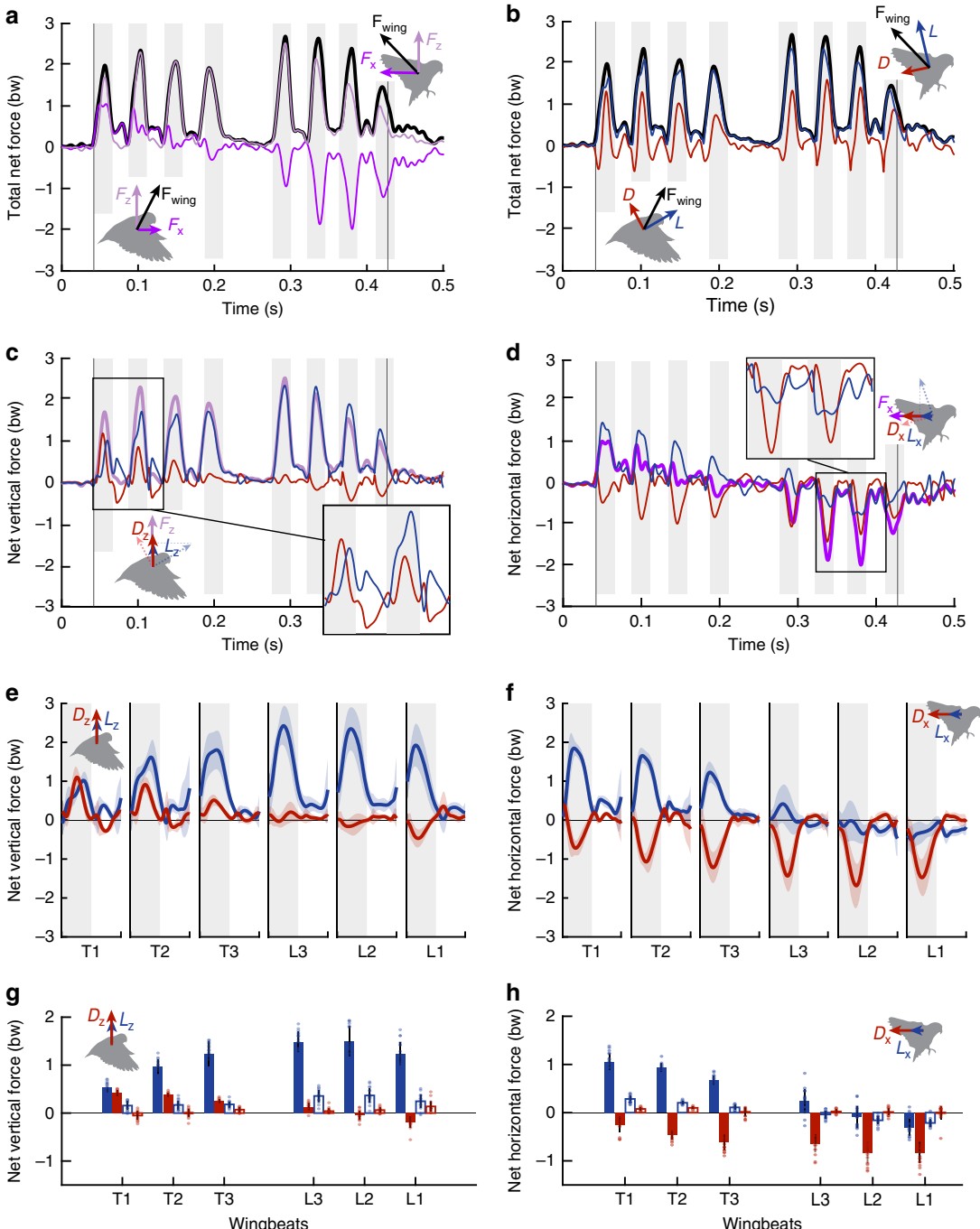

**Fig. 3** Lift can do more than lift, and drag may not be a drag. **a** The net force from both wings (black) is recovered by the vectoral sum of the net horizontal (purple) and vertical forces (light purple) measured in the AFP. Forces are normalized by bodyweight bw. Gray-shaded regions show downstrokes, vertical lines show when perch toe-off and touchdown occur. **b** While net lift (blue) forces tend to be greater than net drag (red) forces, both components contribute significantly to the total net force. **c** The nearly vertical stroke plane after takeoff results in comparable vertical force contributions from both drag and lift during takeoff wingbeats (inset). During later wingbeats, vertical forces are predominately generated through lift. **d** Lift first pitches forward to accelerate the bird and counter drag during takeoff. During landing, lift pitches backward to augment braking forces generated by drag (inset). **a–d** correspond to the same example flight from Fig. 2. Net forces are omitted from insets in (**c**, **d**) to enable better comparison between lift and drag contributions. **e, f** Time-resolved mean ± s.d. force traces ($N = 5$ birds, $n = 4$ flights per bird) show the contributions of lift and drag to the net vertical (**e**) and horizontal (**f**) forces during the first three wingbeats after takeoff (T1, T2, T3) and the final three full wingbeats before landing (L3, L3, L1). **g, h** Stroke-averaged net forces during takeoff and landing wingbeats show how aerodynamic force is generated primarily during downstrokes (filled bars) rather than upstrokes (open bars). **g** While downstroke lift provides the majority of the total weight support, drag also contributes significant vertical force during initial downstrokes, as does upstroke lift throughout each flight. **h** Horizontal accelerating forces are derived primarily from downstroke lift, while braking forces are derived from downstroke drag. Lift also provides braking force during the final two wingbeats. Bars and error bars show mean ± s.d. for $N = 5$ birds, $n = 4$ flights per bird (overlaid dots show individual flights). Source data are provided as a Source Data file.

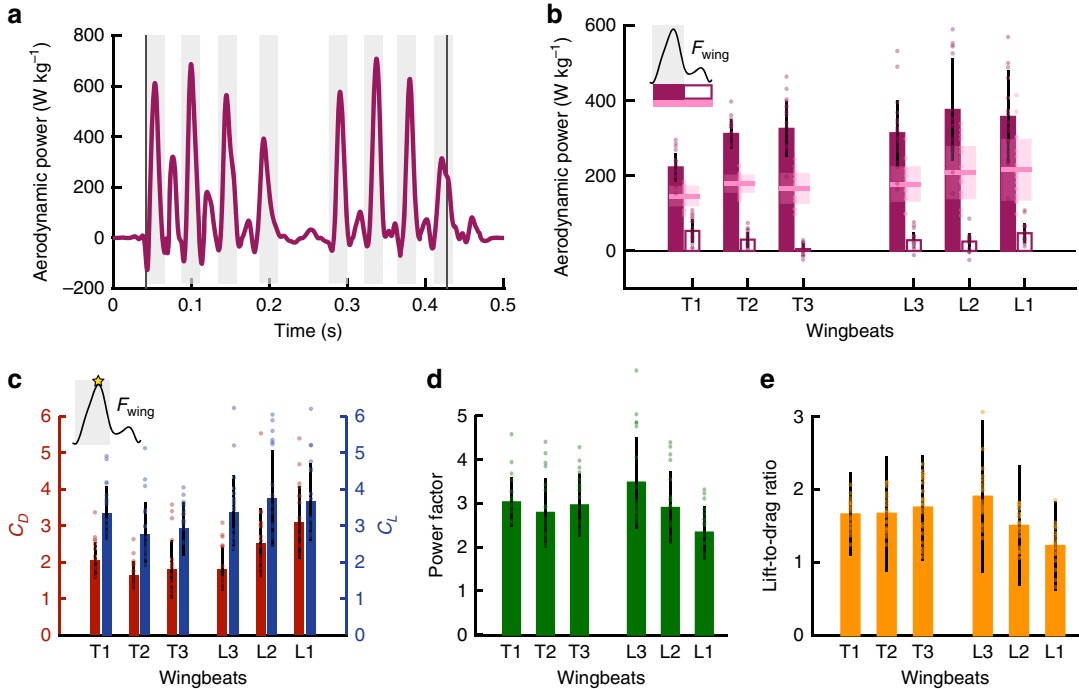

**Fig. 4** Relatively high drag forces provide an expensive way to enhance total aerodynamic force. **a** High aerodynamic power requirements (normalized by pectoralis muscle mass) result from large drag forces during downstrokes and initial upstrokes, as shown for the same representative flight from previous figures. **b** Downstroke-averaged power requirements (filled bars) are much larger than upstroke averages (open bars), especially during mid-flight. Light pink lines and shaded regions show mean ± s.d. for the full wingbeat. **c** High instantaneous drag coefficients $C_D$ (red) and lift coefficients $C_L$ (blue) during maximum net force generation suggest the presence of a leading-edge vortex. **d** The high lift coefficients contribute to relatively high instantaneous power factors ($C_L^{1.5}/C_D$), (**e**) while high drag coefficients result in relatively low lift-to-drag ratios ($C_L/C_D$). All bar plots show mean ± s.d. for $N = 5$ birds, $n = 4$ flights per bird (overlaid dots show individual flights). Source data are provided as a Source Data file.

were based on pectoralis strain measurements calibrated with traditional pull-tests (50–180 W kg$^{-1}$[36,37]) or calibrated based on quasi-steady aerodynamic models (80–250 W kg$^{-1}$[38,39]). Although these previous values did not account for power output from the supracoracoideus during the upstroke, our results (Fig. 4b) indicate that upstroke power output would indeed be less during slow flight. The wingbeat-averaged power outputs derived here from direct force measurements also agree reasonably well with our previous estimate (160 W kg$^{-1}$) based on quasi-static modeling for a parrotlet[23]. The power underestimate of the quasi-steady model was to be expected based on the finding that the quasi-steady model underpredicts the drag of a flapping wing at high angles-of-attack[33].

By beating their wings at a high angle-of-attack of up to 60° during takeoff[7], parrotlets are able to increase both lift and drag to maximize the resultant force vector. Deetjen et al.[7] predicted that these high angles-of-attack would enable parrotlets to use drag for weight support and lift for thrust based on a quasi-steady lift and drag model. We now confirm this prediction in vivo (Fig. 3g, h) and find that the parrotlets use both high lift and high drag coefficients (Fig. 4c). The lift coefficients of around 3 can only be explained by an attached leading-edge vortex (LEV), previously quantitively visualized in hovering hummingbirds[40] and slow-flying fly-catchers[41]. Leading-edge vortices not only increase lift[42], they also increase drag. Increased drag has previously been viewed as a penalty or a necessary side-effect for maintaining muscle efficacy[34,43], but we now see that the added drag may actually be desirable during takeoff. Aerodynamic force produced by a wing is equal to $\frac{1}{2}C_F\rho V^2 S$, where $C_F$ is the coefficient of force, $\rho$ is the air density, $V$ is the wing velocity, and $S$ is the wing surface area. Air density is constant and wing area is already maximized during the downstroke, which means that

either $C_F$ or wing velocity must increase to increase the total aerodynamic force. However, increasing wing velocity would require increasing wing amplitude further, which is morphologically constrained, or increasing muscle contraction frequency, which is restricted to a narrow range for optimizing muscle efficacy[44]. Frequency is further constrained when inertial energy losses are to be mitigated by elastic recoil, which requires operating close to a resonant frequency of elastic storage[45,46]. Considering these constraints, the most parsimonious (remaining) solution is to increase $C_F$, which can be done by generating a LEV. The LEV increases both lift and drag coefficients, which together maximize the total aerodynamic force coefficient $C_F$ at high angles-of-attack.

Generating a high total aerodynamic force coefficient comes at the cost of a lower power factor (PF), a measure of how much lift is generated with a unit of power[34]. Lentink and Dickinson[34] showed that model fly wings flapping at a Reynolds number of 14,000, close to parrotlets, attain a maximal stroke-averaged PF of ~1.6 when the wing has an angle-of-attack around 20° midstroke. The parrotlet power factors during mid-downstroke of 2.8 ± 0.8 (Fig. 4d) show the parrotlet wing is effective, despite operating at much higher angles-of-attack. This is due to the better aerodynamic performance of the wing: although drag coefficients during these flights are high, lift coefficients remain even higher (Fig. 4c), so the corresponding PF are relatively large. This helps bound the high power needed for slow flight. Parrotlets limit the power needed to fly short distances further by utilizing effective takeoff angles with their legs, which enables them to utilize their long jump power to cover more distance[23]. Our study now shows that parrotlets further limit energetic expenditure during landing by using lift, which does not cost aerodynamic power, to supplement braking forces (L2, L1; Fig. 3f, h). However, the use of

drag for weight support (T1, T2, T3; Fig. 3e, g) shows they invest significant power in maximizing the aerodynamic force they can generate with their flapping wings during takeoff.

In addition to high power requirements, the large drag forces on each wing result in low lift-to-drag ratios. The lift-to-drag ratio is commonly used as a measure of flight efficacy[8,47], particularly for birds flying at cruise velocities with a vertical stroke plane, which corresponds to advance ratios (the ratio of the forward wingtip velocity $V_x$ to the wingtip velocity component in the stroke plane) of ~0.6–1.3[5,48]. The parrotlets in this study flew at lower speeds and with inclined stroke planes, reaching advance ratios $J$ of only 0.2–0.3. Their mid-downstroke lift-to-drag ratios, which averaged $1.57 \pm 0.35$ across takeoff and landing wingbeats, (Fig. 4e), are lower than what has been reported for other birds flying at similar advance ratios ($J \approx 0.3$, $C_L/C_D = 8$–$10^{12,36,49}$). This discrepancy results from the use of body velocity, rather than wing velocity, to define the directions of lift and drag; in these previous studies, lift is considered a vertical force that counters bodyweight while drag acts horizontally to counter forward thrust[41,49]. When lift and drag are instead based on the effective velocity at the wing, we find that the lift-to-drag ratios for these other birds decrease to values similar to the parrotlets' ($C_L/C_D \approx 1.5$, see Methods). As Wang, 2004 suggested for hovering insect flight based on 2D simulations, unless the stroke plane is horizontal, the lift-to-drag ratio may not provide an accurate reflection of what forces are useful for a given flight. We now see how this applies to birds in vivo, particularly when drag provides useful aerodynamic forces to support bodyweight during takeoff (Fig. 3e, g) and assist with braking before landing (Fig. 3f, h).

Inclined stroke planes have also been observed in the flight of other birds[1,8–15] and bats[1,16], which suggests that similar uses of lift and drag may be more widespread across flapping animal flight than generally appreciated. Increasing both lift and drag appears to be particularly helpful for generating sufficient weight support when forward flight speed is low, the stroke amplitude and flapping frequency are limited, or wing loading is high. The utility of using an inclined stroke plane may thus be increased in birds with relatively high wing loadings: juvenile birds[50], seabirds that both fly and swim underwater[47], and primitive birds like the hoatzin[51]. Ground birds with robust but low-endurance flight muscles[52] could similarly benefit from being able to repurpose high lift and high drag to take off and land during their short burst flights.

The surprising utility of drag in birds also suggests that the wings of avian precursors could have provided useful aerodynamic forces, even if they were not capable of generating significant lift. Modern birds have asymmetric, lift-generating primary feathers, and many can spread the tips of their primary feathers to create a slotted wingtip configuration for reducing lift-induced drag[53]. On the other hand, many avian precursors were limited to symmetric, drag-based feathers[19,20], or had other morphological constraints that limited their lift-generating capabilities according to paleontological studies[54,55]. While this may have been prohibitive for enabling sustained flapping flight, their wings could have still employed drag forces to provide limited weight support over short distances, just as the parrotlets do during their takeoff wingbeats. In fact, the weight support supplied by drag during their takeoff wingbeats (T1, T2; Fig. 3e) is similar to the total weight support from partial wingbeats that they use to extend long jumps between closely positioned perches[23]. Our finding that drag provides significant weight support therefore lends further support to the idea that foraging proto-birds could have gradually developed their flapping flight abilities by extending long jumps with partial wingbeats[23]. Even proto-birds with symmetric feathers would have been able to generate sufficient weight support for increasing their jump range by repurposing drag forces with an inclined stroke plane.

## Methods

**Birds and training**. We trained five Pacific parrotlets (*F. coelestis*; $30.7 \pm 2.6$ g, three male and two female, 20 Hz wingbeat frequency, $22.0 \pm 1.5$ cm mid-downstroke wingspan) to fly between two perches in the aerodynamic force platform (Fig. 1b). The parrotlets were trained using habituation and positive reinforcement (via millet seed rewards) to fly from the takeoff perch to the landing perch when cued by the trainer's finger or a target stick. Multiple perch-to-perch flights were made by each parrotlet before experimental data were recorded. We recorded 4 flights from each bird for a total of 20 flights across birds. Bird cages are enriched, and birds receive water and food ad libitum. All training and experimental procedures were approved by Stanford's Administrative Panel on Laboratory Animal Care, and no animals were sacrificed for this study.

**Force measurements**. Net vertical and horizontal forces were directly measured using a new aerodynamic force platform (AFP) shown in Fig. 1b, of which we detailed the governing equations elsewhere[21]. The top, bottom, front, and back sides of the AFP flight chamber (1 m length × 1 m height × 0.6 width) are formed by carbon fiber sandwich panels, each attached in a statically determined manner to three Nano 43 sensors (six-axis, SI-9–9.125 calibration; ATI Industrial Automation) sampling at 2000 Hz with a resolution of 2 mN. The sensors are directly attached to stiff support structures that rest statically determined on the ground. Horizontal and vertical aerodynamic forces are determined by summing the corresponding normal and shear forces measured by the four force plates. The side walls of the AFP are made up of clear acrylic sheets for visual access. Although these walls are not instrumented, we can assume that the net lateral force is negligible during forward flight, because lateral forces generated by the right and left wings have to cancel to fly straight. We also added a takeoff perch 10 cm from the back plate and a landing perch 10 cm from the front plate, both at a height halfway between the top and bottom plates. Each perch is constructed from a 5/8″-diameter (1.59 cm) wooden dowel rigidly attached to a carbon fiber beam which extends out of the AFP through small windows in the acrylic side wall. These carbon fiber beams are each instrumented with three ATI Nano 43 sensors (2000 Hz sample rate, 2 mN resolution) which are fixed to mechanically isolated support structures that also rest statically determined on the ground. By combining forces measured by the perches and force plates, we can recover the complete transfer of vertical and horizontal impulse for the first time, to our knowledge. These flights start and end at rest, so we expect that the total vertical impulse imparted by the legs and wings should equal full bodyweight (bw) support[23] and that the total horizontal impulse should equal zero. By integrating the forces from takeoff to landing, we measured a vertical impulse of $-1.01 \pm 0.06$ bw-s and horizontal impulse of $0.07 \pm 0.02$ bw-s, or roughly a 1% error in the vertical direction and 7% error in the horizontal direction. The integration of net aerodynamic force includes the aerodynamic force on the perches. Without perches we find an impulse of $-0.99 \pm 0.05$ bw-s in the vertical direction and $0.06 \pm 0.02$ bw-s in the horizontal direction. All force measurements were filtered using an eighth-order Butterworth filter with a cutoff frequency of 80 Hz for the plates and 40 Hz for the perches, which had a lower natural frequency (>44 Hz) than the force plates (>92 Hz).

**Kinematics**. The body and wingbeat kinematics were captured using five high-speed cameras (three Phantom Miro M310s, one R-311, and one LC310, 1280 × 800 resolution, 1000 fps), synchronized with each other and the force sensors. To enable accurate 3D kinematics, four cameras were positioned at various heights and angles along one side of the AFP, and the fifth camera faced an acrylic window that was built into the front force plate. The cameras were calibrated using the DLT software[56] with an average DLT error <1%. The position of the bird's left eye, left wingtip (distal end of the 10th primary feather), left shoulder, and most distal tip of the tail were manually tracked using the DLT software, and then the data were digitally filtered using Eilers' smoother[57]. The position of a bird's center of mass was estimated based on a weighted sum of the eye and tail positions (69% eye and 31% tail, based on mass distributions measured from two previously sacrificed lovebirds). We used the velocity of the shoulder $v_{shoulder}$ and wingtip $v_{wingtip}$ to estimate the wing's velocity at its second moment of area $r_2$ and at its third moment of area $r_3$ for calculating lift, drag, and aerodynamic power (see "Calculating lift, drag, and power" below). Assuming a linear velocity distribution along a wing with radius $R$, the wing velocity at $r_2$ becomes $v_2 = r_2/R * (v_{tip} - v_{shoulder}) + v_{shoulder}$ and the wing velocity at $r_3$ becomes $v_3 = r_3/R * (v_{tip} - v_{shoulder}) + v_{shoulder}$. To combine our kinematic measurements (recorded at 1000 Hz) and force measurements (sampled at 2000 Hz), we up-sampled the kinematics to match the force data using cubic spline interpolation.

The average Reynolds number for these flights was calculated as $\overline{Re} = \frac{\bar{c}\bar{v}_{wing}}{\nu} \approx 10{,}000$, where $\bar{c}$ is the mean chord length (single wing surface area divided by span, $0.0039$ m$^2$/$0.10$ m $= 0.039$ m), $\bar{v}_{wing}$ is the average wing velocity at $r_2$ ($4.16 \pm 0.25$ m s$^{-1}$), and $\nu$ is the kinematic viscosity of air ($1.6 \times 10^{-5}$ kg m$^{-1}$ s$^{-1}$) at the pressure (100.4 kPa) and temperature range (295–300 K) measured during these flights, based on Sutherland's equation[58] and the ideal gas law. The maximum

Reynolds number ($Re_{max} \approx 20{,}000$) is based on the maximum velocity at the wingtip ($8.15 \pm 0.78$ m s$^{-1}$).

During postprocessing of the data, we found that the shoulder joint was difficult to track as precisely in some video frames for four out of the five birds in the experiment; lighting conditions combined with the very light, pale blue color of these four birds resulted in overexposed videos. Fortunately, by using multiple camera views for tracking and comparing against videos of the fifth bird, which is a darker blue color and much less subject to overexposure, we expect that this limitation was not a significant source of error in our results. We estimate that tracking of the shoulder joint would have deviated at most 5 mm from the actual anatomical position. For a fully extended wing (i.e., during mid-downstroke), this would yield an error in the wing radius direction of at most 3°, and only up to 6° for a retracted (mid-upstroke) wing. The effect of shoulder tracking error on $\mathbf{v}_2$, the wing velocity at $r_2$, is further constrained because $\mathbf{v}_2$ is determined primarily by the much larger wingtip velocity rather than the body/shoulder velocity (Fig. 1c). We also compared the components of $\mathbf{v}_2$ (Supplementary Fig. 1) and wing radius direction (Supplementary Fig. 2) for the darker blue bird against the pale birds, and this confirmed the small differences are primarily due to biological variation.

**Body and tail lift and drag**. We estimate expected body and tail contributions based on lift and drag coefficients reported in the literature. Combined body and tail drag coefficients reported for passerines and swifts range from 0.2 to 0.4[59,60]. Starlings also have tail lift coefficients of about 0.4 across a range of spread angles[61]. We therefore assume a body and tail drag coefficient $C_d = 0.4$ for the parrotlets. The total body and tail drag is $D = \frac{1}{2}\rho \bar{V}^2 S_b C_d = 0.001$ N (<1% bodyweight), where $\rho$ is the density of air (1.2 kg m$^{-3}$) at the pressure (100.4 kPa) and temperature range (295–300 K) measured during these flights, $\bar{V}$ is the average flight speed (1.73 m s$^{-1}$), and $S_b$ is the body frontal area. We determined the body frontal area $S_b = 0.0019$ m$^2$ from a frame extracted from the frontal camera view of a parrotlet bounding during one of its flights in the AFP. This gave a more conservative (larger) estimate than the area given by the scaling equation used in previous studies[60,62], $S_b = 0.0129$ m$^{0.614} = 0.0015$ m$^2$. Lift-to-drag ratios for zebra finch and pigeon tails at slow flight speeds are ~1[63,64], so we expect the total lift from a parrotlet's tail to be as low as its drag. Based on these negligible force contributions from the body and tail, we attribute the total force measured by the AFP to the wings during the parrotlets' flapping flight.

We do measure a small amount of force during bounds that a few of the parrotlets made mid-flight ($F_{x,bound} = -0.04 \pm 0.03$ bw, $F_{z,bound} = 0.14 \pm 0.03$ bw; $N = 3$ birds, $n = 6$ bounds total). However, these forces were low compared with the takeoff and landing wingbeat forces analyzed in this study ($F_{x,downstroke} = 0.70 \pm 0.42$ bw, $F_{x,upstroke} = 0.22 \pm 0.14$ bw, $F_{z,downstroke} = 1.49 \pm 0.39$ bw, $F_{z,upstroke} = 0.34 \pm 0.17$ bw; $N = 5$ birds, $n = 20$ flights). The low forces make these measurements more subject to noise. The vertical forces, in particular, are significantly larger than expected given that the average vertical body accelerations derived from our high-speed videos are near gravity during these bounds ($-9.3 \pm 2.6$ m s$^{-2}$). The lift ($13 \pm 3\%$ bw) and drag ($6 \pm 4\%$ bw) measured during these bounds result in unrealistically high lift coefficients ($9.4 \pm 1.5$) and drag coefficients ($3.9 \pm 2.5$); low flight speeds make these coefficients particularly sensitive to noise (e.g., a 1% bw change in force corresponds to a force coefficient difference of at least 0.7). We therefore believe that the estimates described above based on lift and drag coefficients from the literature should be more representative of aerodynamic forces from a parrotlet's body and tail. Zebra finch, which are also small perching birds, similarly produce <1% bw in body lift and drag during slow flight at similar advance ratios[13].

**Aerodynamic forces from pressure vs. friction**. We make the assumption that aerodynamic forces from a wing act perpendicular to its radius, because these forces are largely comprised of pressure forces at the Reynolds number (ratio of pressure to viscous forces) associated with these flights (~10$^4$; see "Kinematics" above) and at the high angles-of-attack at which the parrotlets beat their wings[7]. Pressure-based mechanisms for lift generation on the wing include leading-edge vortices[34,42]. Aerodynamic drag on the wing results from induced drag, which is also pressure-based, as well as profile drag, which includes both pressure drag (from boundary layer development and separation) and skin friction drag (from boundary layer friction)[25]. However, we expect that the friction component of profile drag does not significantly alter the orientation of the force on a wing, because it contributes relatively little to the total aerodynamic force; assuming a friction drag coefficient of $C_f = 0.02$[65,66], we estimate a maximum friction contribution on each wing of only $D_f = \frac{1}{2}\rho C_f \bar{v}_{wing}^2 S = 0.01$ bw (with air density $\rho = 1.2$ kg m$^{-3}$, maximum wingtip velocity $\bar{v}_{wing} = 8.15$ ms$^{-1}$, and single wing area[23] $S = 0.0039$ m$^2$).

**Calculating lift, drag, and power**. As described in the main text, we calculated lift $L$, drag $D$, and aerodynamic power $P_{aero}$ using Eqs. (1) and (2). In order to do so, we first calculated the effective distances along the wing where the total aerodynamic force and moment on a wing act. Respectively, these are the wing's second and third moments of area, $r_2(t) = \sqrt{\frac{1}{S(t)}\int_0^{R(t)} r^2 c(r)dr}$ and

$r_3(t) = \sqrt[3]{\frac{1}{S(t)}\int_0^{R(t)} r^3 c(r)dr}$, where $R$ is the wing radius, $S$ is the single wing area, $c$

($r$) is the local wing chord at radius $r$, and $dr$ is the infinitesimal wingspan[67]. We determined the chord distribution by segmenting an image of a fully extended parrotlet wing into 20 equally spaced wing strips. The single wing area at each instant was then estimated using $r_3(t) = \int_{R_{max}-R(t)}^{R_{max}} c(r)dr$, where $R_{max}$ is the radius of the fully extended wing, and $R(t)$ is based on the distance from a bird's left shoulder to its left wingtip at each instant in time. We could then define unit vectors corresponding to the direction of $\mathbf{v}_2$, the wing's velocity at $r_2$:

$$\widehat{\mathbf{v}}_2 := \begin{pmatrix} e_{v_2,x} \\ e_{v_2,y} \\ e_{v_2,z} \end{pmatrix} := \frac{1}{|v_2|}\begin{pmatrix} v_{2,x} \\ v_{2,y} \\ v_{2,z} \end{pmatrix}, \tag{3}$$

and a unit vector aligned with the wing radius, $\widehat{\mathbf{r}}$, which points from the bird's left shoulder to its left wingtip.

Assuming the wings beat symmetrically, then $F_x$ and $F_z$ from each wing are equal to half of the net $F_x$ and $F_z$ measured in the AFP. The directions of lift and drag can be derived from our measured 3D kinematics. Drag, by definition, acts opposite the direction of wing velocity:

$$\begin{pmatrix} e_{D,x} \\ e_{D,y} \\ e_{D,z} \end{pmatrix} = -\begin{pmatrix} e_{v_2,x} \\ e_{v_2,y} \\ e_{v_2,z} \end{pmatrix}. \tag{4}$$

Lift, by definition, is orthogonal to the wing velocity. Assuming that the total force, which is predominately a pressure force, acts perpendicular to the wing radius, then lift will also be perpendicular to the wing radius. The direction of lift on the left wing can therefore be determined as:

$$\begin{pmatrix} e_{L,x} \\ e_{L,y} \\ e_{L,z} \end{pmatrix} = \frac{\widehat{\mathbf{v}}_2 \times \widehat{\mathbf{r}}}{|\widehat{\mathbf{v}}_2 \times \widehat{\mathbf{r}}|} \tag{5}$$

Substituting Eqs. (4) and (5) into (1), we arrive at a system of three equations to solve for the three unknowns: the lateral force $F_y$, drag $D$, and lift $L$.

We can then calculate the total aerodynamic power as $P_{aero} = 2(\mathbf{F}_{wing} \bullet \mathbf{v}_3)$, where $\mathbf{v}_3$ is the wing's velocity at $r_3$. We note that this power cost only considers the rate of external work done on the air, and therefore does not include other sources of metabolic or mechanical power, such as inertial power (see ref. [23] for inertial power estimates during short foraging flights). To compare power requirements with other bird studies, we normalized $P_{aero}$ by the flight muscle (pectoralis) mass of a parrotlet. We estimated pectoralis mass as 16% of body mass based on measurements from three sacrificed parrotlets not used in this study ($16.0 \pm 0.8\%$ body mass; $N = 3$).

The calculated lateral force and aerodynamic power become sensitive to error when the vertical and horizontal components of drag and lift are near zero or are parallel or antiparallel, which often occurs during stroke reversal. Our lab member Marc Deetjen developed a solution for this issue that we summarize here for reference and together will publish in detail elsewhere. Mathematically, this sensitivity arises because the solution for deriving $F_y$, $D$, and $L$ requires taking the inverse of the matrix $\mathbf{E} = \begin{pmatrix} e_{D,x} & e_{L,x} \\ e_{D,z} & e_{L,z} \end{pmatrix}$, so when $\mathbf{E}$ is nearly singular (non-invertible), the calculated lateral force can reach unrealistically high values. We therefore apply a regularizing weighting scheme for $F_y$ based on the determinant of $\mathbf{E}$, which approaches zero when $\mathbf{E}$ is near singular. We multiply the value of $F_y$ at each instant in time by the following weight:

$$W = 1 - \max\left\{0, \min\left\{1, \frac{\log|\det \mathbf{E}| - \log c_1}{\log c_2 - \log c_1}\right\}\right\}, \tag{6}$$

where $c_1$ and $c_2$ are tunable constants that determine the strength of this regularization technique. The calculated value of $F_y$ is left unchanged when the determinant of $\mathbf{E}$ (det $\mathbf{E}$) is sufficiently large ($W = 1$ when det $\mathbf{E} > c_1$). As $\mathbf{E}$ approaches singularity, the solution becomes more sensitive, so $F_y$ is attenuated more; $W$ approaches 0 as det $\mathbf{E}$ approaches $c_2$, and $W = 0$ when det $\mathbf{E} < c_2$. We found that setting $c_1 = 0.35$ and $c_2 = 0.05$ enables us to eliminate the large spikes in lateral force with no effect on mid-downstroke lift and drag values (Fig. 4c–e) and minimal effects on power (Fig. 4a, b); applying this weighting scheme changes downstroke and wingbeat averages by <2%, while changes in upstroke averages vary from <1 to 16%.

**Lift and drag coefficients and power factor calculation**. The lift $C_L$ and drag $C_D$ coefficients of the wings were calculated as $C_L = \frac{L}{\frac{1}{2}\rho S v_2^2}$ and $C_D = \frac{D}{\frac{1}{2}\rho S v_2^2}$, where $L$ is the total lift on a single wing, $D$ is the total drag on a single wing, $\rho$ is the density of air (1.2 kg m$^{-3}$) at the pressure (100.4 kPa) and temperature range (295–300 K) measured during these flights, $S$ is the area of a single wing (as described above in "Calculating lift, drag, and power"), and $v_2$ is the magnitude of the wing's velocity at $r_2$. We show $C_L$ and $C_D$ (Fig. 4c) at the instant in time when the net aerodynamic force is maximized ($56 \pm 14\%$ downstroke), because these are the most relevant for evaluating the presence of a leading-edge vortex on the wing. Our confidence in the $C_L$ and $C_D$ calculations is also highest at this instant, because they become highly

sensitive to noise when aerodynamic forces are low and/or when wing velocity is low. We then used these same values to calculate the mid-downstroke power factor PF during each wingbeat (Fig. 4d), where $PF = C_L^{1.5}/C_D$.

**Lift-to-drag ratio comparisons.** Our mid-downstroke lift-to-drag ratios (Fig. 4e) were also calculated using the single wing lift and drag coefficients. We note that they could be equivalently derived based on the instantaneous single wing lift and drag during peak net force, $\frac{C_L}{C_D} = \frac{L/.5\rho S v_2^2}{D/.5\rho S v_2^2} = \frac{L}{D}$.

In order to fairly compare our lift-to-drag ratios to those published in the literature, we limited our comparisons to flights made by generalist birds at similar advance ratio (~0.3)[12,36]. We then converted the lift-to-drag ratios published in these studies based on body velocity direction ($L'/D'$) to lift-to-drag ratios based on the wing velocity direction at $r_2$ ($L/D$). We assumed that the $r_2$ distances along the wings of these other birds (a pied flycatcher and a magpie) are proportionally similar to that of a parrotlet at mid-downstroke ($r_2/R = 0.53$). We then estimated the horizontal and vertical velocity components of the wing due to its flapping motion as $v_x = \frac{r_2}{R} v_{wingtip}\cos(\Phi)$ and $v_z = \frac{r_2}{R} v_{wingtip}\sin(\Phi)$, where $v_{wingtip}$ is the average wingtip velocity and $\Phi$ is the stroke plane angle. Then defining the angle formed between the effective wing velocity at $r_2$ and the horizontal plane as $\phi = \tan^{-1}\frac{v_z}{v_x + U_\infty}$, where $U_\infty$ is the forward body velocity, we can estimate lift and drag as $L = L'\cos(\phi) - D'\sin(\phi)$ and $D = D'\cos(\phi) + L'\sin(\phi)$. This estimate assumes that $L$ and $D$ act primarily in the $x$–$z$ plane (that lateral force on each wing is negligible). While this is not necessarily true throughout the wingbeat, it is a reasonable assumption during mid-downstroke when the wing radius is nearly horizontal, which is when we are making our comparison. The new lift-to-drag ratio can then be calculated as:

$$\frac{L}{D} = \frac{\frac{L'}{D'}\cos(\phi) - \sin(\phi)}{\cos(\phi) + \frac{L'}{D'}\sin(\phi)}. \tag{7}$$

**Reporting summary.** Further information on research design is available in the Nature Research Reporting Summary linked to this article.

## Data availability

The source data underlying Figs 3g, h and 4b–e are provided as a Source Data file. All other data sets generated and analyzed during the current study are available from the authors on reasonable request.

## Code availability

All Matlab code used for postprocessing the data is available from the authors on reasonable request.

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

## Acknowledgements

We thank Marc Deetjen for his help in developing the lift, drag, and power calculations used in this study. This work was supported by NSF Faculty Early Career Development (CAREER) Award 1552419. D.D.C. was supported by a Stanford Graduate Fellowship and a National Defense Science and Engineering Graduate Fellowship.

## Author contributions

D.D.C. collected and analyzed data. D.D.C. and D.L. designed the study, interpreted the findings, and wrote the paper.

## Competing interests

The authors declare no competing interests.
