## [Peer Review File · Nature Communications]

Reviewers' Comments:

Reviewer #1:

Remarks to the Author:

L29: In static wings - ignores the energy required to overcome inertia in flapping wings.

Legend, Figure 1: The figure is pretty small, so it would help if was more clearly stated. The blue L is total (to me, "net" implies something has been subtracted total lift) lift; the dashed vertical component (grey) opposes W, and the solid grey horizontal component T opposes the dashed red horizontal component of total drag D.

L107: Given that this reasonable assumption is key to the inferences made, a bit more on the reasons would be welcome. For example, how might the substantial portion of the lift generated by leading edge vorticity affect this assumption? Further, induced drag is a pressure force; profile drag on the wing is frictional. For all the tremendous quantification here, "predominantly" is a little unsatisfying.

L301: Was maximizing lift to drag during slow flight a paradigm? For most species, they spend so little time in slow flight that efficiency is irrelevant. They must simply be effective, by whatever means. This study does a nice job describing magnitude of the available means, but I'm not sure that for birds anyone ever reasonably thought that L/D was that critical.

L313: "...[increased] in birds.." "Elevated" denotes position; not to be confused with wing position.

L326, L335 "...leveraged..." I find the overuse of this word, away from its physical definition, unsettling; because the authors are here writing of Newtonian physics, its use at least unsettling; at worst, confusing. "employed" or your original "repurposed".

A terrific and thorough quantification of slow flight. The integration of the perch data with the body impulse is a particularly nice and understated touch.

Reviewer #2:

Remarks to the Author:

The authors investigated how birds use the lift and drag reoriented by adjusting the stroke plane angle for their take-off and landing. They utilized the DLT-based high-speed photogrammetry synchronized with the aerodynamic force platform to measure the wing kinematics and the aerodynamic force simultaneously. The aerodynamic forces in the global frame were decomposed into the forces with respect to the wing motion that is the rigorous definition of the lift and drag. Through the analyses, the authors discovered that the lift can be used for the thrust or braking force as well as the weight support, and the drag can be used for the weight support as well as the braking force.

The work is based on the methods established well by the authors and is undoubtedly of high quality, but I have a concern over the novelty of the study. The decomposition into the lift and drag is based on the wing motion. The quantitative estimate of force and power is the strength of the study, but, without them, we can predict the role of lift and drag in global frame and power by just looking at the wing motion and the angle of attack. Therefore, the discussion on the direction of lift and drag in the earlier studies (such as refs. 11 and 15) are valid as far as the definitions of the lift and drag are valid. It is not clear to me if the quantitative measurement performed in this study offer new insight into the bird flight. At least, I think that the utility of the drag shown in this study is not "surprising" since it has been suggested in previous studies.

Another (minor) concern is the estimate of the power. I guess that the aerodynamic power does not contain the power to rotate the wings? The effect of the wing rotation (around the spanwise axis) may be small, but the negative power during the stroke reversal may be due to the lack of the rotational power in addition to the "local flow field during wing-wake interactions." The power in this study is based on the wing motion and aerodynamic forces, but it does not contain the power to overcome the

wing inertia. It is difficult to estimate the inertial power accurately (since it requires the detailed wing shape), but a simple analysis to compare the inertial power with the aerodynamic power would be helpful to estimate the total power consumption in bird flight.

We much appreciate the positive evaluation of our work by both Reviewers and the helpful review comments we received that enabled us to further improve our manuscript. We provide a point-by-point response below explaining all changes, which we highlighted in blue in the manuscript. We believe that these revisions should fully resolve all of the comments made by the Reviewers. The two key changes are (1) the quantification of expected skin-friction drag to better support our assumption that aerodynamic forces on the wing are pressure-dominated, and (2) we clarified our description of how we approximate and calculate aerodynamic power, both in words and with an added equation in the main text. Both changes are included in the main text and detailed in our Methods section. To simplify the review, we copy and pasted the key changes in our point-by-point responses below. Finally, we further clarified our text as recommended by Reviewer #1.

Reviewers' comments:

Reviewer #1 (Remarks to the Author):

L29: In static wings - ignores the energy required to overcome inertia in flapping wings.

>> Thank you for pointing this out, we forgot to clarify we are discussing external work on the air here. We now clarify by stating:

“In the body frame, the **external work exerted on the air to generate** net lift is zero, because **lift** acts perpendicular to the average body flight velocity and, therefore, does not oppose flight.”

We do agree inertia is a factor in the birds' overall energetic balance, but it isn't a factor in the *aerodynamic* power needed to generate lift that we discuss here. We clarified this by explicitly writing that we consider the “external work exerted on the air” in our study, the aerodynamic power.

Legend, Figure 1: The figure is pretty small, so it would help if was more clearly stated. The blue L is total (to me, “net” implies something has been subtracted total lift) lift; the dashed vertical component (grey) opposes W, and the solid grey horizontal component T opposes the dashed red horizontal component of total drag D.

>> Complied; thank you for pointing this out, we clarified the caption as follows: “**During steady forward flight, total lift (dashed blue arrow)** counters bodyweight **W (solid green arrow)** in the vertical direction, and **total drag D (dashed red arrow)** is countered by net thrust **T (solid dark grey arrow)** in the horizontal direction of body velocity (yellow). **However, during slow flapping flight the total lift L (solid blue arrow) and total drag D (solid red arrow)** vectors generated by an individual wing are directed differently, **because** wing velocity \mathbf{v}_{wing} does not align with body velocity \mathbf{v}_{body} , as shown for a bird's first downstroke after takeoff.”

L107: Given that this reasonable assumption is key to the inferences made, a bit more on the reasons would be welcome. For example, how might the substantial portion of the lift generated by leading edge vorticity affect this assumption? Further, induced drag is a pressure force; profile drag on the wing is frictional. For all the tremendous quantification here, “predominantly” is a little unsatisfying.

>> Complied; the assumption that aerodynamic force acts perpendicular to the wing radius is reasonable because pressure forces (including forces from leading edge vorticity and induced drag) dominate over viscous/friction forces at the Reynolds numbers of these flights (10,000-20,000). Additionally, profile drag includes both pressure and friction drag [1], and at the high

[1] Drela, 2014. *MIT Press*.

angles-of-attack used by the parrotlets [2], the pressure component dominates over the friction [1]. We now clarify why this assumption is reasonable in the main text as:

“we make the reasonable assumption that F_{wing} acts perpendicular to the wing radius r because it is predominately a pressure force at the Reynolds numbers associated with these flights ($\sim 10^4$). This is particularly true for the high angles-of-attack used by the parrotlets⁷, for which pressure-based profile drag dominates over friction-based profile drag²⁵. We estimate that skin friction only reaches a force magnitude of at most 1% bodyweight on each wing (see Methods), and therefore does not significantly alter the direction of the total aerodynamic force on the wing.”

We also added the following section in our Methods:

Aerodynamic forces from pressure vs. friction

We make the assumption that aerodynamic forces from a wing act perpendicular to its radius, because these forces are largely comprised of pressure forces at the Reynolds number (ratio of pressure to viscous forces) associated with these flights ($\sim 10^4$; see *Kinematics*) and at the high angles-of-attack at which the parrotlets beat their wings⁷. Pressure-based mechanisms for lift generation on the wing include leading edge vortices^{34,43}. Aerodynamic drag on the wing results from induced drag, which is also pressure-based, as well as profile drag, which includes both pressure drag (from boundary layer development and separation) and skin friction drag (from boundary layer friction)²⁵. However, we expect that the friction component of profile drag does not significantly alter the orientation of the force on a wing, because it contributes relatively little to the total aerodynamic force; assuming a friction drag coefficient of $C_f = 0.02$ ^{68,69}, we estimate a maximum friction contribution on each wing of only $D_f = \frac{1}{2} \rho C_f \bar{v}_{\text{wing}}^2 S = 0.01 \text{ bw}$ (with air density $\rho = 1.2 \text{ kg/m}^3$, maximum wingtip velocity $\bar{v}_{\text{wing}} = 8.15 \text{ m/s}$, and single wing area²³ $S = 0.0039 \text{ m}^2$).

L301: Was maximizing lift to drag during slow flight a paradigm? For most species, they spend so little time in slow flight that efficiency is irrelevant. They must simply be effective, by whatever means. This study does a nice job describing magnitude of the available means, but I'm not sure that for birds anyone ever reasonably thought that L/D was that critical.

>> We agree – our intended meaning was that maximizing lift to drag is often emphasized for cruising flight (not for slow flight), but we have removed this sentence to avoid confusion.

L313: “...[increased] in birds..” “Elevated” denotes position; not to be confused with wing position.

>> Complied, we now use “increased” instead of “elevated”.

L326, L335 “...leveraged...” I find the overuse of this word, away from its physical definition, unsettling; because the authors are here writing of Newtonian physics, its use at least unsettling; at worst, confusing. “employed” or your original “repurposed”.

>> Complied; we have replaced all instances of “leveraged” as recommended:

“...which enables them to ~~leverage~~ utilize their long jump power to cover more distance...”

“...their wings could have still ~~leveraged~~ employed drag forces to provide limited weight support over short distances...”

“Even proto-birds with symmetric feathers would have been able to generate sufficient weight support for increasing their jump range by ~~leveraging~~ repurposing drag forces with an inclined stroke plane.”

A terrific and thorough quantification of slow flight. The integration of the perch data with the body

[2] Deetjen et al., 2017. *Journal of Experimental Biology*.

impulse is a particularly nice and understated touch.

>> Thank you for this positive evaluation!

Reviewer #2 (Remarks to the Author):

The authors investigated how birds use the lift and drag reoriented by adjusting the stroke plane angle for their take-off and landing. They utilized the DLT-based high-speed photogrammetry synchronized with the aerodynamic force platform to measure the wing kinematics and the aerodynamic force simultaneously. The aerodynamic forces in the global frame were decomposed into the forces with respect to the wing motion that is the rigorous definition of the lift and drag. Through the analyses, the authors discovered that the lift can be used for the thrust or braking force as well as the weight support, and the drag can be used for the weight support as well as the braking force.

>> Thank you for this nice summary of our paper.

The work is based on the methods established well by the authors and is undoubtedly of high quality, but I have a concern over the novelty of the study.

>> We respectfully disagree that our study is not novel; we do not know of any other journal paper that reports the direct measurement of wingbeat-resolved horizontal and vertical aerodynamic forces in flapping animal flight from take-off to landing *in vivo*. Further, this is the first time lift and drag have been derived from direct aerodynamic force measurements during *in vivo* flight. We presented these novelties relatively understated in the manuscript, so we understand that this may have been overlooked. We hope we have now clarified that the direct measurement of these forces is a significant contribution that helps us to evaluate earlier published ideas that were based on indirect inferences.

The decomposition into the lift and drag is based on the wing motion. The quantitative estimate of force and power is the strength of the study, but, without them, we can predict the role of lift and drag in global frame and power by just looking at the wing motion and the angle of attack. Therefore, the discussion on the direction of lift and drag in the earlier studies (such as refs. 11 and 15) are valid as far as the definitions of the lift and drag are valid. It is not clear to me if the quantitative measurement performed in this study offer new insight into the bird flight. At least, I think that the utility of the drag shown in this study is not “surprising” since it has been suggested in previous studies.

>> Thank you for recognizing that the quantitative estimate of force and power is a strength of our study. While we agree and also state in the text that the utility of drag has been suggested in previous studies, its relative contribution to either weight support or forward thrust has never been quantified *in vivo* for bird flight. Whereas earlier measurements of the wing velocity vector were able to give the lift and drag directions, they did not give the lift and drag magnitude – for that the horizontal and vertical forces need to be measured, which we present for the first time. The suggestion that the force vector magnitude can be determined based on angle of attack is incorrect; the lift-drag curve of a flapping bird wing has never been measured *in vivo*, so the *in vivo* relationship between lift, drag, and angle of attack is unknown in birds. There do exist such relationships for prepared wings of bird cadavers that have been spun around with a wing

spinner (e.g. [3,4,5,6]), but those recordings are not *in vivo* and do not replicate the *in vivo* unsteady wingbeat kinematics. It is unknown to what degree these quasi-steady cadaver studies match the actual *in vivo* lift and drag forces since those have never been measured directly before.

In fact, it is not possible to derive the *in vivo* magnitude or ratio of lift and drag without our direct force measurements from our new aerodynamic force platform; without both horizontal and vertical force measurements, the *in vivo* lift and drag cannot be determined. We show this using Equation 1 from the manuscript:

$$\mathbf{F}_{\text{wing}} = \begin{pmatrix} F_x \\ F_y \\ F_z \end{pmatrix} = D \begin{pmatrix} e_{D,x} \\ e_{D,y} \\ e_{D,z} \end{pmatrix} + L \begin{pmatrix} e_{L,x} \\ e_{L,y} \\ e_{L,z} \end{pmatrix}.$$

Wing kinematics alone would only recover the directions of lift and drag, yielding 3 equations for 5 unknowns (D, L, F_x, F_y, F_z). Even our 1D (vertical force) measurements from our previous aerodynamic force platforms [7,8] would not be sufficient for fully constraining the system (3 equations for 4 unknowns). Only by combining our measurements of both horizontal and vertical forces with 3D wing kinematics are we now able to recover the magnitudes of lift and drag for the first time, which are needed to assess the utility and repurposing of lift and drag for forward thrust or weight support.

In summary, no previous studies combine velocity with force measurements as we do, so our study is the first direct and *in vivo* quantification of lift magnitude, drag magnitude, and lift to drag ratios in bird flight. We were therefore able to determine, for the first time, how birds use lift and drag during foraging flights. For instance, no other studies have suggested that drag can support nearly half of a bird's bodyweight during takeoff, or that lift acts as a mechanism for augmenting braking forces during landing, which has critical implications for the power and energy costs associated with these short flights.

To avoid the reasonable confusion that resulted in this comment, we now write in our introduction:

"It would not be possible to solve this governing set of equations with kinematics alone; only by combining our kinematic measurements with our *in vivo* measurements of the net vertical force (\$F_z\$ ) and horizontal force (\$F_x\$ ) are we able to solve for the magnitudes of lift and drag. We were thus able to quantify the role of lift and drag throughout each flight from takeoff to landing for, to our knowledge, the first time."

Another (minor) concern is the estimate of the power. I guess that the aerodynamic power does not contain the power to rotate the wings? The effect of the wing rotation (around the spanwise axis) may be small, but the negative power during the stroke reversal may be due to the lack of the rotational power in addition to the "local flow field during wing-wake interactions."

>> Thank you for this suggestion. We agree that the rotational power is likely small for the parrotlets' wingbeat kinematics, as described in the literature [9,10]. Thus, although total aerodynamic power includes both translational and rotational power, our study focuses on

[3] Usherwood et al., 2002. *Journal of Experimental Biology*

[4] Kruyt et al., 2014. *Journal of the Royal Society Interface*

[5] Altshuler et al., 2004. *Journal of Zoology*

[6] Dial et al., 2012. *Journal of Experimental Biology*

[7] Chin and Lentink, 2017. *Science Advances*.

[8] Ingersoll and Lentink, 2018. *Science Advances*.

[9] Dickinson et al., 1999. *Science*.

[10] Sane and Dickinson, 2001. *Journal of Experimental Biology*.

translational work and assumes rotational power can be neglected, as many previous studies have done [11,12,13]. We now clarify this assumption in the text as follows:

“It is also possible that wing rotation effects may account for some of the negative power, but calculating rotational power would require quantification of net torques that have never been measured before *in vivo*. However, compared to the translational aerodynamic power requirements that we do measure, we expect rotational power requirements to be relatively low, especially given the parrotlets’ large stroke amplitudes (142 ± 9 deg) (Sane and Dickinson, 2001; Dickinson et al., 1999). We therefore assume that the total aerodynamic power P_{aero} can be well approximated as described above, based on the translational component of the aerodynamic power:

$$P_{\text{aero}} = P_{\text{aero,trans}} + P_{\text{aero,rot}} = 2(\mathbf{F}_{\text{wing}} \cdot \mathbf{v} + \mathbf{T}_{\text{wing}} \cdot \boldsymbol{\omega}) \approx 2(\mathbf{F}_{\text{wing}} \cdot \mathbf{v}),$$

where \mathbf{T}_{wing} is the aerodynamic torque on the wing, and $\boldsymbol{\omega}$ is the angular velocity of the wing.”

The power in this study is based on the wing motion and aerodynamic forces, but it does not contain the power to overcome the wing inertia. It is difficult to estimate the inertial power accurately (since it requires the detailed wing shape), but a simple analysis to compare the inertial power with the aerodynamic power would be helpful to estimate the total power consumption in bird flight.

>> Thank you for pointing this out. We forgot to clarify that our analysis of aerodynamic power is only considering external work on the air, for which inertia plays no role. The total power consumption in bird flight would also include other contributions, including metabolic efficiency, which we do not measure and are outside the scope of our study of aerodynamic forces and aerodynamic power (rate of external aerodynamic work). To clarify this in the main text, we added the following:

“The aerodynamic power, which we define as the rate of external work exerted on the air, is calculated as $P_{\text{aero}} = 2(\mathbf{F}_{\text{wing}} \cdot \mathbf{v}_3)$, where \mathbf{v}_3 is the wing’s velocity at its 3rd moment of area r_3 (see Methods).”

We now also further clarify this in the Methods as follows:

“We can then calculate the total aerodynamic power as $P_{\text{aero}} = 2(\mathbf{F}_{\text{wing}} \cdot \mathbf{v}_3)$, where \mathbf{v}_3 is the wing’s velocity at r_3 . We note that this power cost only considers the rate of external work done on the air, and therefore does not include other sources of metabolic or mechanical power, such as inertial power (see ²³ for inertial power estimates during short foraging flights).”

[11] Tobalske et al., 2003. *Nature*.

[12] Berg and Biewener, 2008. *Journal of Experimental Biology*.

[13] Pennycuik, 1968. *Journal of Experimental Biology*.

Reviewers' Comments:

Reviewer #1:

Remarks to the Author:

The changes to the manuscript are appropriate – in particular, the clarifications regarding the decomposition of lift and drag, and the treatment of inertial power. There's a lot going on here, and the more plainly you can describe it (e.g., your simple summary of the study's novelty), the better. Very nice work –